# BP–ANN Model Coupled with Particle Swarm Optimization for the Efficient Prediction of 2-Chlorophenol Removal in an Electro-Oxidation System

**DOI:** 10.3390/ijerph16142454

**Published:** 2019-07-10

**Authors:** Yu Mei, Jiaqian Yang, Yin Lu, Feilin Hao, Dongmei Xu, Hua Pan, Jiade Wang

**Affiliations:** 1College of Environment, Zhejiang University of Technology, Hangzhou 310032, China; 2College of Biological and Environmental Engineering, Zhejiang Shuren University, Hangzhou 310005, China

**Keywords:** BP–ANN, PSO–ANN, electro-oxidation

## Abstract

Electro-oxidation is an effective approach for the removal of 2-chlorophenol from wastewater. The modeling of the electrochemical process plays an important role in improving the efficiency of electrochemical treatment and increasing our understanding of electrochemical treatment without increasing the cost. The backpropagation artificial neural network (BP–ANN) model was applied to predict chemical oxygen demand (COD) removal efficiency and total energy consumption (TEC). Current density, pH, supporting electrolyte concentration, and oxidation–reduction potential (ORP) were used as input parameters in the 2-chlorophenol synthetic wastewater model. Prediction accuracy was increased by using particle swarm optimization coupled with BP–ANN to optimize weight and threshold values. The particle swarm optimization BP–ANN (PSO–BP–ANN) for the efficient prediction of COD removal efficiency and TEC for testing data showed high correlation coefficient of 0.99 and 0.9944 and a mean square error of 0.0015526 and 0.0023456. The weight matrix analysis indicated that the correlation of the five input parameters was a current density of 18.85%, an initial pH 21.11%, an electrolyte concentration 19.69%, an oxidation time of 21.30%, and an ORP of 19.05%. The analysis of removal kinetics indicated that oxidation–reduction potential (ORP) is closely correlated with the chemical oxygen demand (COD) and total energy consumption (TEC) of the electro-oxidation degradation of 2-chlorophenol in wastewater.

## 1. Introduction

Wastewater produced by various industrial processes contains large quantities of chlorophenol compounds, which are highly toxic and resistant to biological degradation [1]. The compound 2-chlorophenol is a typical chlorophenol compound that is listed as a priority pollutant by the Environmental Protection Agency, given its carcinogenic properties [2,3]. Electro-oxidation, an effective technology that does not require the use of extra reagents, is commonly used to remove chlorophenol compounds from wastewater because of its high efficiency, rapid reaction rate, and environmental friendliness [4,5]. However, the energy cost of the electro-oxidation process limits its application [6].

The establishment of appropriate models for electro-oxidation is essential given the complexity of this process. Modeling of the electrochemical process plays an important role in improving the efficiency of electrochemical treatment and a further understanding of electrochemical treatment without increasing the cost. Empirical models and semi-empirical models, such as pseudo-first-order kinetics [7], pseudo-second-order kinetics [8], a computational fluid dynamics (CFD) model, and response surface methodology (RSM) model, are usually established for the prediction of electrochemical process behaviors. Bu et al. [9] established the kinetic model of the degradation of oxcarbazepine (OXC) using electrochemically-activated persulfate (EC/PS) based on two assumptions. Conventional mathematical or mechanistic models can be used to predict the final state of the system only under given circumstances [10]. Wang et al. [11] calculated the velocity distribution and turbulence distribution of a new type of tubular plunger flow reactor by CFD. CFD can reveal the mass transfer process and mechanism of an electrochemical reactor, but it is still affected by grid mass, transfer mode, and calculation [12,13]. Song et al. [14] optimized the electrochemical simultaneous removal of the ammonia nitrogen process using RSM, which showed a good prediction. The main disadvantage of RSM is that it cannot effectively improve approximation accuracy, even with an increase in the number of sample points. Electrochemistry is a complex non-linear process, and it is difficult to explain it clearly through traditional empirical and semi-empirical modeling.

In contrast to traditional mathematical models, scholars have done some research on the non-linear prediction model of the electrochemical process. Artificial Neural Networks (ANNs) do not require the modeling of a detailed mathematical formulation of a system and have been used to determine complex relationships between input and output data [15]. Daneshvar et al. [16] established an ANN model for the decolorization process of dyeing wastewater by electroflocculation. This model can predict the color removal rate under different experimental conditions. Researchers pointed out that ANN has good prospects for the prediction of complex systems [17,18]. Belkacem et al. [19] applied a backpropagation artificial neural network (BP–ANN) prediction of oxytetracycline removal in an electro-oxidation system, which chose 14 nodes from the hidden layer, the LM (Levenberg-Marquardt)algorithm, the logsig transfer function of the hidden layer, and the purelin transfer function of the output layer. However, the researchers did not verify the reliability of the network or compare the algorithms and transfer functions on the network. Moreover, BP–ANN easily falls into the local minimum and has a poor global convergence rate [20]. The further optimization of the BP–ANN has also attracted growing attention [21]. Particle swarm optimization (PSO) is an algorithm that simulates the foraging behavior of birds [22]. Khajeh and coworkers [23] integrated PSO in a BP–ANN model for the specification of optimal initial weights and threshold values by updating generations to avoid the local minimum and achieve global convergence quickly and correctly.

Establishing an efficient and reliable ANN model for predicting the behavior of electrochemical oxidation processes can reduce energy cost and is a fundamental step toward their control. The input parameters of ANN network are one of the key factors in establishing an ANN network. Oxidation–reduction potential (ORP) has been employed as an integrated indicator in various fields to describe the redox characteristic of any given chemical reaction system [24]. ORP has a good relationship with the chemical oxygen demand (COD) of electro-oxidation [25]. Wang and coworkers [26] constructed a model of the multiparameter linear relationship between ORP and Qsp (specific electrical charge) and between a COD and Cl^−1^ concentration to reflect quantitatively the effect of the current density, Cl^−1^ concentration, pollutant load, and reaction time on the electro-oxidation system. Basha et al. [27] built a BP–ANN model to predict the effect of electro-oxidation on COD removal, but ORP was not considered in the input parameters.

In this study, PSO–BP–ANN models were constructed to predict the COD removal efficiency and total energy consumption (TEC) of electro-oxidation. ORP was used as one of the input parameters. First, BP–ANN and the selection of the number of hidden layers and training algorithm were discussed in detail. Then, the PSO algorithm was used to optimize the weight and threshold of BP–ANN and identify the optimal parameters of the PSO algorithm. Experimental values were compared with output variables predicted by PSO–BP–ANN. The importance of each input variable was determined.

## 2. Materials and Methods

### 2.1. Data Set

All electro-oxidation experiments were conducted with a 3 L-capacity laboratory-scale plate cell with a circulating tank. The used datasets were obtained from a previous study [25]. A total of 190 experimental runs (Table A1) were performed in the galvanostatic mode under a current density of 8 mA cm^−2^ to 25 mA cm^−2^, an original pH of 3 to 11, an electrolyte concentration of 0.05mol L^−1^ to 0.12 mol L^−1^, a reaction time of 0 h to 2 h, and ORP values of −68 mV to 500 mV, as shown in Table 1.

During the Electro-oxidation, an ORP (SX-630, Sanxin, China) and a pH (SX711, Sanxin, China) probe were installed in the electrolysis bath for online monitoring of ORP/pH. COD was determined according to Chinese standard HJ/T 399-2007 with slight modifications. The solution was measured at a wavelength of 440 nm using a UV-visible spectrophotometer (UV-2910, Hitachi, Japan).

A specific electrical charge (*Q_sp_*, Ah L^−1^) was calculated by using the following equation [26]:(1)$$Q_{sp}=j\cdot A\cdot \frac{t}{V}$$
where *j* is current density (A cm^−2^), *A* is the effective area of the electrode (cm^2^), *V* is the effective volume of the plate cell (L), and *t* is the reaction time during the electro-oxidation process (h).

TEC (kWh m^−3^) was calculated in a previous study as follows [28]:(2)$$TEC=Q_{sp}\cdot U$$
where *Q_sp_* is a specific electrical charge, and *U* (V) is the cell voltage.

### 2.2. BP–ANN Coupled with PSO

ANNs have different architectures. The ANN used in this study has three layers: an input layer that receives electro-oxidation information, a hidden layer that processes information, and an output layer that calculates COD removal and TEC results [29]. During BP learning, the actual outputs are compared with the target values to derive error signals, which are propagated backward by layers to adjust the weights in all lower layers [30]. The architecture of a neural network and the BP algorithm is presented in Figure 1.

The flowchart of BP–ANN coupled with PSO is shown in Figure 2. The ANN model was developed using MATLAB R2016a software. A total of 190 runs of the electro-oxidation process data were applied to develop the models for the prediction of COD removal efficiency and TEC. The available data were divided into training, validation, and testing subsets, of which 80% (152) were randomly selected for network training, 10% (19) were used for validation, and 10% (19) were applied to test network accuracy. Current density, original pH, electrolyte concentration, oxidation time, and ORP were used as five input parameters, and COD removal efficiency and TEC were considered as the two output.

Two prediction score metrics, the coefficient of correlation (R^2^), and mean square error (MSE), were computed using the following equations to evaluate the fitting and prediction accuracy of the constructed models [31]:(3)$$R^{2}=\frac{\sum_{i=1}^{n}\left(f_{\exp,i}-F_{\exp})(f_{ANN,i}-F_{ANN}\right)}{\sqrt{\sum_{i=1}^{n}\left({\left(f_{\exp,i}-F_{\exp}\right)}^{2}{\left(f_{ANN,i}-F_{ANN}\right)}^{2}\right)}}$$
(4)$$MSE=\frac{\sum_{i=1}^{n}{\left(f_{\exp,i}-f_{ANN,i}\right)}^{2}}{n}$$
where $F_{\exp}=\frac{1}{n}\sum_{i=1}^{n}f_{\exp,i}$, $F_{ANN}=\frac{1}{n}\sum_{i=1}^{n}f_{ANN,i}$, *n* is the number of samples used for modeling, *f*_exp_ is the experimental value, and *f_ANN_* is the network-predicted value.

## 3. Results and Discussion

### 3.1. Removal Kinetics

The apparent reaction rate constants for COD removal were calculated in accordance with Equation (5) [32]:(5)$$\ln[COD_{t}]=\ln[COD_{0}]-Kt$$
where *COD*_0_ and *COD_t_* are the COD values of the initial and final pollutant concentrations (mg L^−1^), respectively; *t* is the electrolysis time (min); and *K* is the apparent reaction rate constant (min^−1^). The apparent reaction rate constants calculated in accordance with Equation (3) for the current densities of 8, 10, 12, 14, 15, 18, 20, and 25 mA cm^−2^ were 0.0072, 0.0107, 0.0118, 0.0160, 0.0202, 0.0212, 0.0224, and 0.0232 min^−1^, respectively. The linear relationship between the logarithmic values of COD and electrolysis time is depicted in Figure 3. Table 2 shows that the correlation coefficient R^2^ of linear fitting was greater than 0.9989. This result indicates that COD removal satisfies the first-order reaction kinetics equation.

Other parameters, such as temperature (T), pH value, and electricity can be obtained when the influent quality and flow rate are held constant in the electrolytic cell. The kinetic constant K is only related to current density (*j*) under the conditions of the original pH of 3 and Na_2_SO_4_ concentration of 0.10 mol L^−1^ [11].

(6)$$K=Mj^{a}$$

The relationship between *K* and *J* can be inferred from Table 2.

(7)$$K=0.0012j^{0.9485}$$

From Equation (5), Equation (7) can be expressed as
(8)$$\ln[COD_{t}]=\ln[COD_{0}]-0.0012j^{0.9485}t$$
which describes the relationship among COD, current density, and oxidation time.

The optimal electro-oxidation conditions were initially determined by considering the effective factors of current density, original pH value, and electrolyte concentration. A COD removal efficiency of 100% was obtained with the optimal operating parameters of a current density of 15 mA cm^−2^, an original pH of 3, and a Na_2_SO_4_ concentration of 0.10 mol L^−1^ at 120 min. The dependencies of the values of COD, ORP, TEC, and Qsp under a current density of 15 mA cm^−2^, an original pH of 3, and a Na_2_SO_4_ concentration of 0.10 mol L^−1^ during electrochemical oxidation are shown in Figure 4. COD removal efficiency, TEC, and Qsp increased with electro-oxidation time. COD removal efficiency, TEC, Qsp, and ORP were 77.9%, 24.2 kWh m^−3^, 1.375 Ah L^−1^, and 383 mV, respectively, when oxidation time was 1 h. The ORP value decreased from 494 mV to 190 mV within 5 min of electrolysis and then increased gradually to 500 mV during degradation.

The typical multiple regression equation showing the relationship among ORP, current density, original pH, Na_2_SO_4_ concentration, reaction time, and COD removal efficiency was obtained as follows:(9)$$\begin{array}{l} COD\%=-0.16276+0.00281j+0.01709pH+1.5595[Na_{2}SO_{4}]\\ +0.00495t+9.766624E-4ORP \end{array}$$

The typical multiple regression equation representing the relationship among influential parameters and TEC was obtained and is shown below:(10)$$\begin{array}{l} TEC=-39.06431+1.97416j+0.2894pH+66.72156[Na_{2}SO_{4}]\\ +0.46082t+0.00664ORP \end{array}$$

The R^2^ values for COD removal efficiency and TEC were 0.8878 and 0.93223, respectively. These values reflect a good correlation among COD, TEC, *j*, pH, *t*, Na_2_SO_4_ concentration, and ORP. ORP values provide a complete indicator of the effect of current density, electrolyte concentration, pH, and reaction time on the performance of the electro-oxidation system. Therefore, the ORP value can be used as an effective controlling factor for the prediction of COD removal efficiency and the TEC of electro-oxidation.

### 3.2. BP–ANN Prediction of 2-Chlorophenol Removal

The tangent sigmoid was selected as the transfer function for the input layer nodes to the hidden layer, and the purelin was selected as the transfer function for the hidden layer nodes to the output layer. All data were normalized within a range of −1 and 1 before being fed to the networks to increase training speed and facilitate modeling and prediction.

In this study, the numbers of input and output nodes were 5 and 2, respectively, and were equal to the numbers of input and output data. The number of neurons has a considerable effect on network performance. For example, the network cannot achieve the desired error if the number of neurons is too small, or overfitting may occur if the number of neurons is too large. Thus, determining the appropriate number of neurons in the hidden layer is necessary. This number can usually be determined by using the following empirical formula in accordance with Hecht–Nielsen’s theorem [33]:(11)$$N_{H}=2N_{i}+1$$
where *N_H_* is the number of hidden neurons, and *N_i_* is the number of input variables, which is 5 in the present work. Equation (11) shows that the node number in the hidden layer was approximately 11. Then, BP networks with different hidden neurons from 6–16 were compared on the basis of the maximization of R^2^ and the minimization of MSE for the testing dataset. Table 3 shows that the BP–ANN that contains 6–16 hidden neurons in the prediction of the electro-oxidation process. The optimal BP–ANN model provided an R^2^ and MSE of 0.9344 and 0.0137232 for COD removal efficiency, respectively, and an R^2^ and MSE of 0.9355 and 0.013127 for TEC, respectively when the hidden neurons were 10. Under the optimal network, BP–ANN in the prediction of COD removal efficiency and TEC and the correlations between the experimental and predicted sets are illustrated in Figure 5. The error range of COD was (−0.058, 0.249) and TEC (−0.079, 0.391). The network performance is good, but the error range shows that the deviation of individual points is large.

The training algorithm also affects the performance of BP networks. A wide variety of training functions with 10 neurons used in the hidden layer was studied to select a good BP network. Table 4 presents the data for R^2^ and MSE under different training functions of BP networks. The Levenberg–Marquardt back propagation (trainlm) training algorithm, which maximized the R^2^ and minimized the MSE of COD removal efficiency and TEC, was identified as the best training function.

### 3.3. Optimization of the Weight and Threshold Value of BP–ANN

The PSO–BP–ANN can be optimized for selection purposes by optimizing (1) swarm size, (2) maximum iteration, (3) cognition coefficient C_1_, and (4) social coefficient C_2_ (Table A2). Table 5 displayed PSO control parameters, R^2^, and training MSE for the testing dataset. The PSO–ANN containing a swarm size of 50, a maximum iteration of 200, C_1_ of 1.5, and C_2_ of 1.5 was selected as the best model for the electrochemical process of interest. The optimal PSO–BP–ANN models provided R^2^ of 0.99 and 0.9944 for COD removal efficiency and TEC, and MSE values of 0.0015526 and 0.0023456, respectively, for the testing dataset. The performance of the optimal PSO–BP–ANN in the prediction of COD removal efficiency and TEC and the correlations between the experimental and predicted sets are illustrated in Figure 6. The PSO–BP–ANN selected for the efficient prediction of 2-Chlorophenol removal in an electro-oxidation system was containing 10 hidden neurons, trainlm training algorithm, swarm size of 50, maximum iteration of 200, C_1_ of 1.5, and C_2_ of 1.5.

### 3.4. Assessment of the Importance of Variables

The weight matrix of the neural net can be used to assess the relative importance of various input variables for output variables [31]. The relative importance of input variables on the value of COD removal efficiency and TEC as calculated by particle swarm optimization BP–ANN (PSO–BP–ANN) is shown in Table 6. Sensitivity analysis indicated order of relative importance the operational parameters on the electro-oxidation as: electrolysis time > pH > electrolyte concentration > ORP > current density. The table indicates that all of the variables have strong effects on COD removal efficiency and TEC. Therefore, none of the variables studied in this work should be neglected in the analysis.

## 4. Conclusions

In this study, the main object is development and construction of novel model that could make efficient prediction of electro-oxidation removal of 2-Chlorophenol on the basis of batch electro-oxidation experiments. The analysis of removal kinetics indicated that ORP was closely correlated with COD removal efficiency and TEC and was one of the important input parameters of PSO–BP–ANN. PSO–BP–ANN was developed through the optimization of the weights and thresholds of BP–ANN. The PSO–BP–ANN that contained 10 hidden neurons, trainlm training algorithm and possessed a swarm size of 50, maximum iteration of 200, C_1_ of 1.5, and C_2_ of 1.5 was identified as the best model for predicting 2-chlorophenol degradation through electro-oxidation. The PSO–BP–ANN model provided accurate predictions and R^2^ of 0.99 and 0.9944 for COD removal efficiency and TEC, and MSE values of 0.0015526 and 0.0023456 respectively for the testing dataset. The weight matrix revealed that the order of relative importance for the operational parameters of the electro-oxidation is: electrolysis time > pH > electrolyte concentration > ORP > current density. For comparative purposes, performance data for the ANN methodology in various electrochemical processes are summarized in Table A3.

## Figures and Tables

**Figure 1 ijerph-16-02454-f001:**
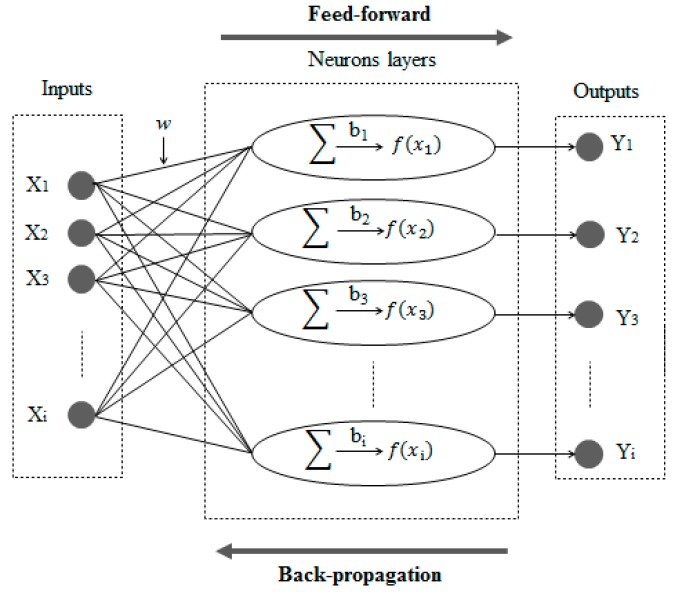
Architecture of an artificial neural network (ANN) and feed-forward back-propagation algorithm.

**Figure 2 ijerph-16-02454-f002:**
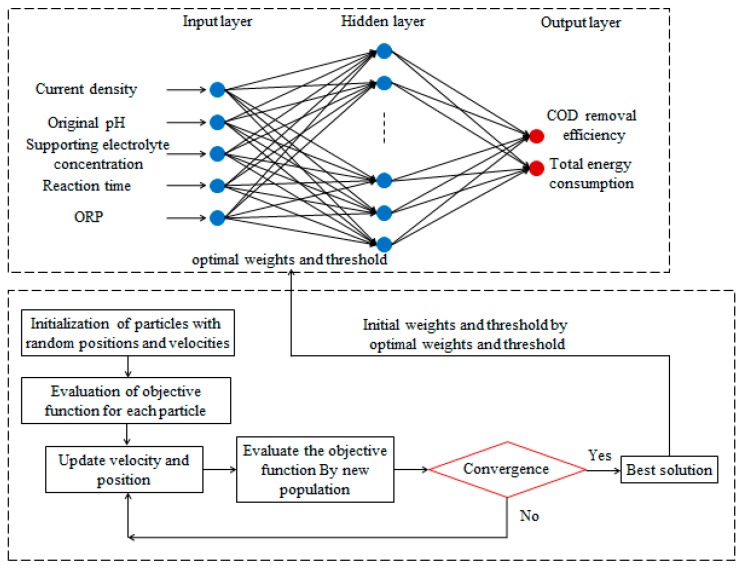
Flowchart of a backpropagation artificial neural network (BP–ANN) combined with particle swarm optimization (PSO).

**Figure 3 ijerph-16-02454-f003:**
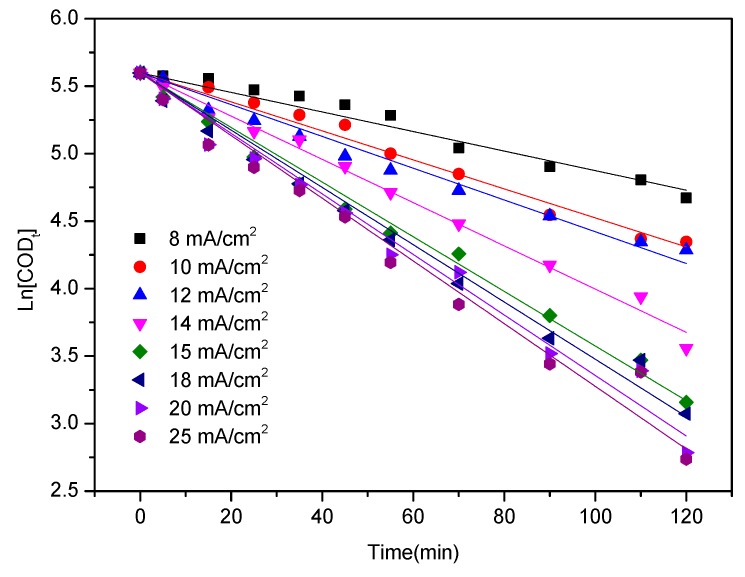
Linear relationship between the logarithmic values of chemical oxygen demand (COD) and electrolysis time.

**Figure 4 ijerph-16-02454-f004:**
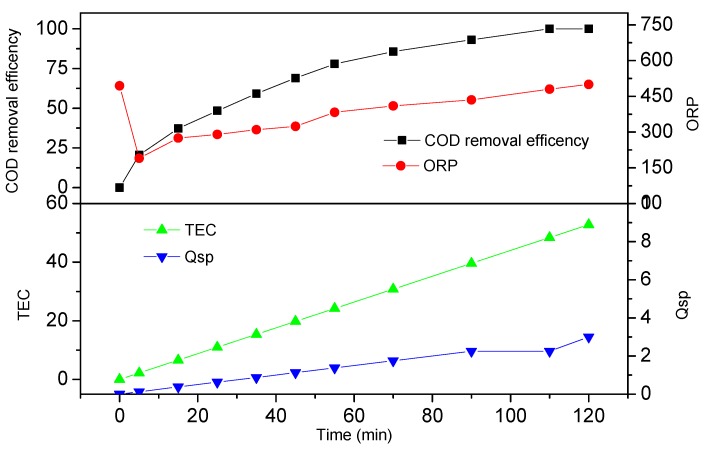
COD removal efficiency, ORP, total energy consumption (TEC), and Qsp under a current density of 15 mA cm^−2^, original pH of 3, and an Na_2_SO_4_ concentration of 0.10 mol L^−1^.

**Figure 5 ijerph-16-02454-f005:**
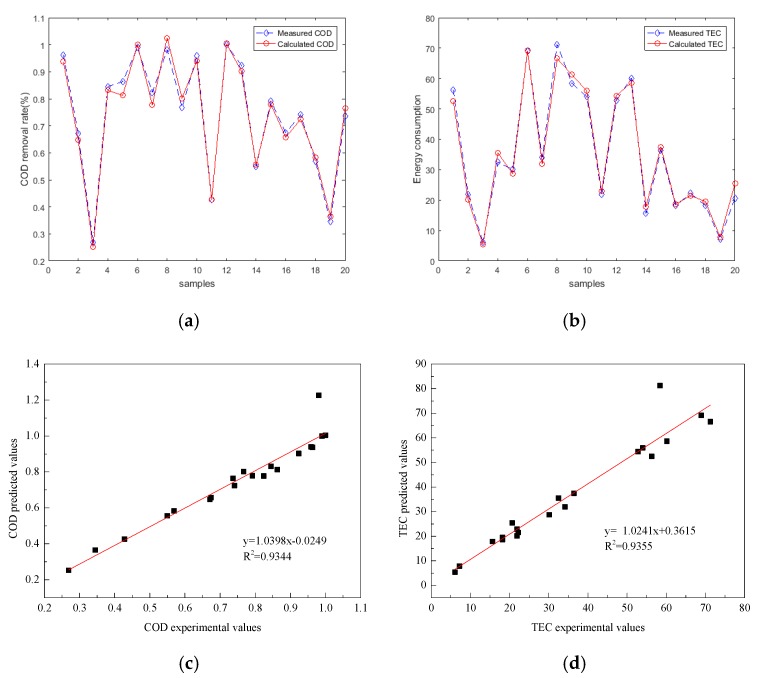
Performance of the BP–ANN predicting COD removal efficiency and TEC between experimental and predicted data sets (COD removal efficiency testing set (**a**), TEC testing set (**b**)); correlations between experimental and predicted set (COD removal efficiency testing set (**c**), TEC testing set (**d**)).

**Figure 6 ijerph-16-02454-f006:**
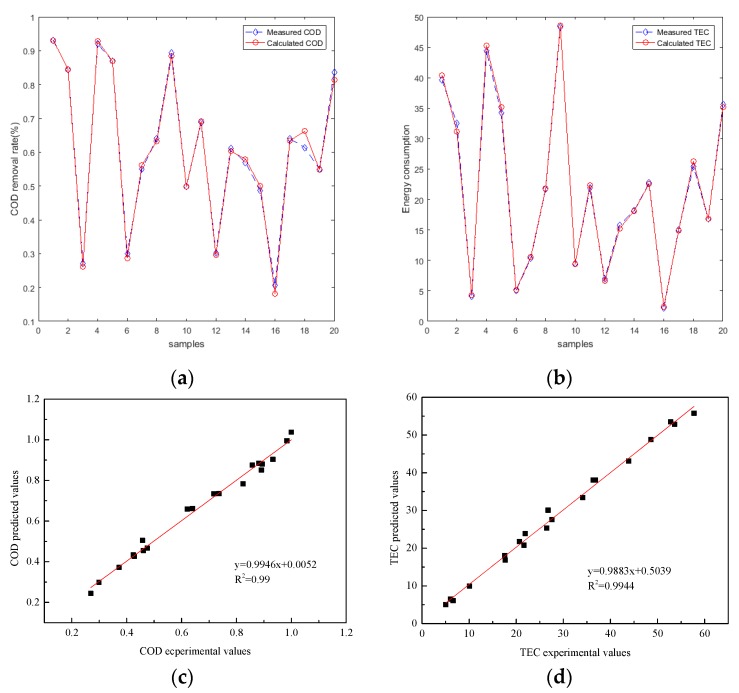
Performance of the particle swarm optimization BP–ANN (PSO–BP–ANN) predicting COD removal efficiency and TEC between experimental and predicted data sets (COD removal efficiency testing set (**a**), TEC testing set (**b**)); correlations between experimental and predicted set (COD removal efficiency testing set (**c**), TEC testing set (**d**)).

**Table 1 ijerph-16-02454-t001:** Experimental conditions. ORP, oxidation–reduction potential.

Run no.	Current Density (mA cm^−2^)	Na_2_SO_4_ Concentration (mol L^−1^)	Initial pH	Electrolysis Time (h)	ORP	Flow Mode
0–190	8–25	0.05–0.12	3–11	0–2	−68–500	continuous

**Table 2 ijerph-16-02454-t002:** K and correlation coefficient values under various current densities.

Current Density *j* (mA cm^−2^)	Regression Line	K (min^−1^)	R^2^
8	Y = 0.00724x + 5.59842	0.0072	0.9999
10	Y = −0.01074x + 5.59842	0.0107	0.9999
12	Y = −0.01177x + 5.59842	0.0118	0.9998
14	Y = −0.01602x + 5.59842	0.0160	0.9998
15	Y = −0.02023x + 5.59842	0.0202	0.9997
18	Y = −0.02121x + 5.59842	0.0212	0.9995
20	Y = −0.02242 + 5.59842	0.0224	0.9992
25	Y = −0.02322 + 5.59842	0.0232	0.9989

**Table 3 ijerph-16-02454-t003:** Evaluation of the prediction performance of the BP–ANN model for the testing dataset.

*N_H_*	COD Removal Efficiency		TEC	
	R^2^	MSE	R^2^	MSE
6	0.9151	0.0155151	0.9277	0.014145
7	0.8741	0.0127321	0.8896	0.013234
8	0.8781	0.0152728	0.9025	0.016566
9	0.9292	0.0149617	0.9148	0.003826
10	0.9344	0.0137232	0.9355	0.013127
11	0.8998	0.0146919	0.9051	0.016887
12	0.8447	0.0165818	0.9077	0.014058
13	0.9032	0.0141709	0.9185	0.013157
14	0.8231	0.0158827	0.893	0.016551
15	0.874	0.0165818	0.8987	0.014344
16	0.8451	0.0153163	0.9021	0.013923

**Table 4 ijerph-16-02454-t004:** Predictions of backpropagation (BP) models with different training algorithms for the testing dataset.

BP–ANN	Training Function	COD Removal Efficiency		TEC	
		R^2^	MSE	R^2^	MSE
Batch training with weight and bias learning rules	trainb	0.86209	0.0134868	0.88977	0.0162386
BFGS quasi-Newton backpropagation	trainbfg	0.90721	0.0161285	0.77684	0.0184532
Bayesian regularization backpropagation	trainbr	0.8426	0.012	0.84645	0.0157329
Unsupervised batch training with weight and bias learning rules	trainbu	0.91427	0.0143475	0.84693	0.0159821
Cyclical order weight/bias training	trainc	0.79387	0.0183421	0.78352	0.0173493
Powell-Beale conjugate gradient backpropagation	traincgb	0.84096	0.0183258	0.81842	0.016399
Fletcher-Reeves conjugate gradient backpropagation	traincgf	0.88913	0.0159525	0.89006	0.0144586
Polak-Ribi’ere conjugate gradient backpropagation	traincgp	0.89724	0.0153866	0.73305	0.0191479
Batch gradient descent	traingd	0.91312	0.016002	0.88845	0.0158414
Gradient descent with adaptive learning rate back propagation	traingda	0.91939	0.0191324	0.88416	0.0159636
Batch gradient descent with momentum	traingdm	0.88482	0.0163147	0.85786	0.0184368
Variable learning rate backpropagation	traingdx	0.91799	0.0143824	0.78431	0.0189369
Levenberg–Marquardt back-propagation	trainlm	0.9344	0.0137232	0.9355	0.013127

**Table 5 ijerph-16-02454-t005:** PSO–ANN with different parameters of the PSO algorithm.

Number of Neurons	Swarm Size	Max Iteration	Cognition Coefficient (C_1_)	Social Coefficient (C_2_)	COD Removal Efficiency		TEC	
					R^2^	MSE	R^2^	MSE
10	10	200	1.5	1.5	0.9528	0.0024367	0.9781	0.0024975
10	30	200	1.5	1.5	0.9783	0.0034865	0.9878	0.0022
10	50	200	1.5	1.5	0.99	0.0015526	0.9944	0.0023456
10	70	200	1.5	1.5	0.976	0.0015874	0.9878	0.0038921
10	100	200	1.5	1.5	0.9736	0.00173	0.9977	0.003281
10	120	200	1.5	1.5	0.98	0.0019062	0.9983	0.0031672
10	50	100	1.5	1.5	0.9852	0.0011566	0.9834	0.0012677
10	50	150	1.5	1.5	0.9695	0.0021488	0.9876	0.001835
10	50	250	1.5	1.5	0.9891	0.0012508	0.9812	0.0033047
10	50	200	0.5	2.5	0.9767	0.0024646	0.9882	0.0026686
10	50	200	1	2	0.9888	0.00179873	0.9891	0.0012586
10	50	200	2	1	0.9874	0.0023016	0.9919	0.0034017

**Table 6 ijerph-16-02454-t006:** Relative importance of input variables on the value of COD removal efficiency and TEC.

Input Variable	Importance (%)
current density	18.85%
original pH	21.11%
electrolyte concentration	19.69%
electro-oxidation time	21.30%
ORP	19.05%
Total	100%

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
