# Peer review of "BP–ANN Model Coupled with Particle Swarm Optimization for the Efficient Prediction of 2-Chlorophenol Removal in an Electro-Oxidation System"

_ijerph, 2019, doi:10.3390/ijerph16142454_

Round 1

Reviewer 1 Report

This article is modeling the electro-chemical oxidation process via artificial neural network using BP–ANN model. The prediction accuracy of model was investigated using particle swarm optimization, which seems to be interesting.

Abstract:

1.       how you prove this statement: chemical oxygen demand (COD) and total energy consumption (TEC) of 14 the electro-oxidation degradation? At least do not start with this statement at the beginning, but at the end of abstract.

2.       Abbreviation of particle swarm optimization (PSO)

3.       Stated that it is synthetic or real effluent. I it is synthetic surely the model has high predication accuracy.

Introduction:

1.       General referencing from 1 to … is not the way to effectively cite some study, at least it should be haphazard to show you completely cover one study.

Material and methods

1.       What is the concentration of contaminants, 2-chlorophenol?

2.       How did you change ORP?

3.       Better to present the results of 190 experiments in supplementary files.

Results:

1.       Do you model the system in the case of Mass transfer control?

2.       Regression line in Table 2 should passes through a set point which is Ln (CODo)

3.       Indicate the reason behind higher performance of EOP under acidic pH (seems that it should be reverse).

4.       Supply training function code in supplementary file

Author Response

Responds to reviewer 1 comments:

Point 1:

Abstract:

1.     How you prove this statement: chemical oxygen demand (COD) and total energy consumption (TEC) of the electro-oxidation degradation? At least do not start with this statement at the beginning, but at the end of abstract.

2.     Abbreviation of particle swarm optimization (PSO)

3.     Stated that it is synthetic or real effluent. I it is synthetic surely the model has high predication accuracy.

Response 1:

1.     Thank you for reviewer’s comment. It has been modified in the article and marked with the yellow line.

2.     It has been modified in the article and marked with the yellow line.

3.     All wastewater in this article is synthetic wastewater. It has been stated in the article and marked with the yellow line.

Point 2:

Introduction:

1.     General referencing from 1 to … is not the way to effectively cite some study, at least it should be haphazard to show you completely cover one study.

Response 2:

1.  Thank you for reviewer’s comment. It has been modified in the article and marked with the yellow line.

Point 3:

Material and methods

1.       What is the concentration of contaminants, 2-chlorophenol?

2.       How did you change ORP?

3.       Better to present the results of 190 experiments in supplementary files.

Response 3:

1.     Thank you for reviewer’s comment. The concentration of 2-chlorophenol is 150 mg L-1.

2.     ORP was a comprehensive indicators in wastewater advanced treatments. It was affected by operating parameters including current density, initial pH and Na2SO4 concentration. ORP changed with the experimental conditions in the article.

3.     I will present the results of 190 experiments in supplementary files as Table S1.

Point 4:

Results:

1.     Do you model the system in the case of Mass transfer control?

2.     Regression line in Table 2 should passes through a set point which is Ln (CODo)

3.     Indicate the reason behind higher performance of EOP under acidic pH (seems that it should be reverse).

4.     Supply training function code in supplementary file

Response 4:

1.     Thank you for reviewer’s comment. Many electro-oxidation processes are controlled by mass transfer. The electrochemical device was contained a reactor, a peristaltic pump and a circulating tank. The 2-CP degradation rate at different flow rate (peristaltic pump with 0, 20, 40, 60, 80, 100 rpm) was calculated before batch experiments. The results showed the degradation rate was increased with flow rate and the optimum flow rate was 266 ml min-1 (100 rpm). The batch experiments were performed under the optimum flow rate to eliminate the impact of mass transfer.

2.     I redraw Fig.3 picture which pass through a set point Ln(COD0). And the new regression line was showed in Table 2 in the article.

3.     The removal efficiency of 2-chlorophenol increased with the initial pH value decrease. The acid-favored was attributed to the difficulty of total oxidation in alkaline condition. The pH value after dichlorination will increase. Considering that the final solution at neutral is propitious, a lower initial pH is required.

4.     I will supply training function code in supplementary file as Table S2.

Reviewer 2 Report

- good research article;

- description of the methods used for the identification of chemical and physical parameters should be provided;

- comparison with the newest results of other authors should be provided;

Author Response

Dear Reviewers:

    I am very grateful to your comments concerning our manuscript entitled      “BP-ANN model coupled with particle swarm optimization for the efficient      prediction of 2-Chlorophenol removal in an electro-oxidation system".       Thank you for the reviews for giving us constructive suggestions.       We have studied comments carefully and have made correction which we       hope meet with approval. Revised portion are marked in yellow background       in the revised article file. The responds to the reviewers’ comments are       as following.

Best regards!

Responds to reviewer 2 comments:

Point 1:

description of the methods used for the identification of chemical and  physical parameters should be provided;

Response 1: Thank you for reviewer’s comment. It has been modified in the article and marked with the yellow line.

Point 2:

comparison with the newest results of other authors should be provided;

1.     Response 2: Thank you for reviewer’s comment. It has been supplied in supplementary file as Table S3.

This manuscript is a resubmission of an earlier submission. The following is a list of the peer review reports and author responses from that submission.